# Differentiating between Nitrogen and Water Deficiency in Irrigated Maize Using a UAV-Based Multi-Spectral Camera

**Taylor Becker [1],\*, Taylor S. Nelsen [1]**, **Michelle Leinfelder-Miles [2] and Mark E. Lundy [1,3]**

1    Department of Plant Sciences, University of California, Davis, CA 95616, USA;
     tsnelsen@ucdavis.edu (T.S.N.); melundy@ucdavis.edu (M.E.L.)
2    University of California Cooperative Extension, San Joaquin County, Stockton, CA 95206, USA;
     mmleinfeldermiles@ucanr.edu
3    Division of Agriculture and Natural Resources, University of California, Davis, CA 95618, USA
\*    Correspondence: tbecker@ucdavis.edu

**Abstract:** The objective of this research was to determine if canopy reflectance measured by an Unmanned Aerial Vehicle (UAV) equipped with a 5-band multi-spectral camera can differentiate between water and nitrogen (N) deficiency in irrigated maize. Crop reflectance was used to generate a Normalized Difference Red Edge (NDRE), Green Leaf Index (GLI), and a Blue Reflectance Index (BRI). These indices were then used in combination to categorize N and water stressed experimental units into a Combined Index (CI) indicating water-stressed, N-stressed, or non-stressed crops. The CI generated at blister (R2) successfully identified 90% of experimental treatments to the correct group but only identified 60% of treatments when generated at the 14th leaf stage (V14). The CI methodology was subsequently applied to two independent site-years where only N deficiency gradients were imposed. The CI was not successful at separating treatments at the validation sites, incorrectly identifying water stress where there was none. Among individual indices investigated, NDRE had the strongest relationship to grain yields ($r^2 = 0.62$, $p < 0.001$) but a weaker linear relationship compared to the CI ($r^2 = 0.68$, $p < 0.005$) where deficit irrigation treatments were imposed. At sites where irrigation was sufficient to meet crop water demand, NDRE ($r^2 = 0.63$, $p < 0.05$) had a stronger relationship to grain yield compared to the CI ($r^2 = 0.41$, $p = 0.31$). This study found that, under narrow cropping system circumstances, N and irrigation-induced differences in maize productivity can be differentiated in-season by a combination of reflectance indices, but that NDRE alone provides superior information under broader contexts.

**Keywords:** maize; nitrogen stress; water stress; UAV; drone; canopy reflectance

## 1. Introduction

Nitrogen (N) is a limiting nutrient in most cropping production systems and accounted for 57% of all applied plant nutrients in the US in 2014 [1]. Of the 12,061,021 metric tons of N used in the US in 2014, maize production alone accounted for 48% of the total [1]. N is a critical macronutrient necessary for maximizing yield in maize cropping systems [2,3]. For N applications, the amount, form and timing of N are the most important management factors affecting grain yield. To a lesser extent, soil type, climate, and cultivar also contribute to grain yields [4,5].

Nutrient applications and irrigation are both key management considerations for improving yield of field crops, including maize [6–8], and mismanagement of these resources can have environmental and economic consequences [9]. It is important to apply the right rate of N fertilizer at the right time, not only to avoid stressed crops, but also to prevent negative environmental impacts. Specifically,

N leaching from agricultural land to ground and surface water has become a problem warranting regulatory action in many regions, including the state of California. It has been shown that crops with the greatest soil N availability can have the greatest losses from the soil due to leaching, but not necessarily the highest yields or significantly greater N uptake [10]. These losses may be preventable with better management and understanding of the quantity and pattern of N uptake for each crop. For example, N uptake rates in maize reach a maximum at a late vegetative growth stage, around the 10th to 14th leaf stages [11]. By the beginning of the reproductive stages, maize has acquired about 70% of the total N it will take up through full maturity [11]. Site and growth stage specific N fertilizer applications can reduce the risk of N loss [5,12]. However, applying only as much fertilizer as is required by the crop to meet yield potential is a challenge in agricultural production due to the interactions of management, the environment, and crop genotype with nitrogen use and allocation. This challenge becomes even more complicated when simultaneously considering crop water use, since N and water resources are co-limiting, meaning the lack of one resource negatively affects the efficient use of the other [13].

Common methods of estimating the N status of a crop include soil testing or destructive plant sampling. These methods are expensive, time consuming, and labor intensive, especially in large fields with high variability. Remote and proximal sensing methods have been developed to overcome these barriers. Remote sensing involves the use of satellites or airborne vehicles to evaluate spatially explicit data while proximal sensing occurs closer to the area of interest and includes handheld or land-based vehicle-mounted devices that measure reflectance. These sensing devices are useful alternatives to plant sampling, and have been used to approximate yield and N uptake in maize [14,15]. One tool used to assess nutrient deficiencies in maize is reflectance measured by crop canopy sensors [16,17]. Multi-spectral reflectance measured by remote sensing devices has allowed agronomists to describe stress across a larger spatial area, capturing more in-field heterogeneity compared to traditional plant sampling. This advance has led to the development of objective decision making tools for N application leading to variable rate N applications in maize [16,18,19]. One remote sensing method of capturing canopy reflectance is the use of Unmanned Aerial Vehicle (UAV) mounted multi-spectral cameras. UAVs, also referred to as drones, have become useful in agriculture for quickly capturing field-scale heterogeneity that is spatially sensitive.

These UAV mounted multi-spectral (typically 5-band) cameras are becoming increasingly available and are commonly used in agricultural research to capture images and monitor crops. The bands and wavelengths commonly measured using a multispectral camera are the red (630–690 nm), green (520–600 nm), blue (450–520 nm), near-infrared (760–900 nm), and red-edge (700–730 nm) [20]. The resulting reflectance data can be used to generate at least 22 different vegetation indices, some of which have been shown to be highly correlated to maize canopy chlorophyll content, biomass production, and overall yields [21–23]. The success of individual reflectance indices in identifying stress is a result of the fact that both N and water stressed plants tend to have a higher reflectance in the visible spectrum and reduced reflectance in the near-infrared region compared to non-stressed plants [24–26]. Among the various indices calculable from the 5 bands available in most commercial multi-spectral cameras, the normalized difference red-edge (NDRE) has been used to approximate crop stress in maize and other crops [27,28]. NDRE is identical in calculation to the normalized difference vegetative index (NDVI) but substitutes an average red-edge band for the average red band used in NVDI [27]. The red-edge region of the visible reflectance spectrum is a region of rapid increase in reflectance of vegetation where chlorophyll does not absorb radiation [17]. NDRE is a superior measurement for detecting canopy cover and chlorophyll content compared to the more commonly used NDVI because NDVI saturates in value at dense canopy cover more so than NDRE [29]. The Green Leaf Index (GLI) is calculated based on reflectance from the blue, green and red bands, which are all within the visible spectrum. GLI is highly correlated to total leaf chlorophyll content, and has long been used to monitor the N sufficiency status of maize [30]. By quantifying visible color differences that are correlated to crop N content, GLI might be useful as an additive factor

to magnify N-based differences between stressed and un-stress crops when used in tandem with a N-sensitive index like NDRE. Finally, blue reflectance has been used to detect differences in chlorophyll or other carotenoid concentrations that are related to N and water stress [31,32]. The blue reflectance may also be useful in separating vegetation from soil, since it is included in the calculation of the Enhanced Vegetation Index (EVI). EVI has been shown to effectively differentiate between soil and vegetation [33], suggesting that blue reflectance may be more sensitive to stress that affects vegetative cover as opposed to color.

Very few studies have explored the use of multispectral imaging to simultaneously identify nitrogen and water stress. Despite reflectance and vegetation indices being commonly used to indicate stress, a complicating factor to the estimation of N stress using multi-spectral images is that crop canopy reflectance is influenced by many factors other than nutrient deficiency. Specifically, differentiating between nutrient and water stress is difficult because water stress increases reflectance in the visible and NIR region, which are regions that are commonly used to identify nutrient deficiency [34]. Expensive hyperspectral and thermal cameras have been used to identify water stress in maize, but price and technical barriers have limited the adoption of these technologies within a management context [35,36]. Therefore, the development of an index able to identify water stress without more expensive equipment would be useful for management as well as research purposes [37]. Although previous studies have reported on the abilities of NDRE, GLI and blue reflectance to indicate stress individually, whether combinations of these indices might result in additive information about crop N and water status is unknown. Because of the interacting effects of N and water deficiency on crop reflectance, it is important to explore the potential differences in reflectance between water and N stressed crops using tools available for general agricultural research and management. NDRE, GLI, and the blue reflectance, are a combination of indices that utilize every reflectance band measured by most commercially-available 5-band cameras.

Therefore, the objective of this study was to use multi-spectral reflectance from UAV-generated aerial images to differentiate nitrogen from water stress and quantify the severity of these types of stress in irrigated maize. We hypothesized that crop reflectance and a vegetation index or combination of indices can be used to identify stress severity and differentiate between nitrogen and water stress across gradients of nitrogen and water deficiencies. This hypothesis was tested by: (1) imposing gradients of N and water deficiency at the field scale; (2) measuring crop N uptake and water use under varying stress types and levels; (3) quantifying the relationship between cropping system outcomes and select vegetation indices using reflectance in the visible and near-infrared regions; and (4) validating the resulting relationship at two additional site-years.

## 2. Materials and Methods

### 2.1. Experimental Set-Up

Field experiments were conducted in 2018 and 2019 at the University of California, Davis Russell Ranch Sustainable Agriculture Facility located in Yolo County, CA (38° 2′32.95″ N, 121°52′31.11″ W) and in 2018 at a grower field in the Sacramento Delta region near Eagle Tree, CA (38°11′21.28″ N, 121°31′50.97″ W). The dominant soil type at Russell Ranch is a Rincon silty clay loam which is considered a mineral soil with 2% organic matter. The dominant soil type at the Delta site is a Rindge mucky silt loam which has much higher organic matter content than the Russell Ranch soil at 17.5% [38]. Another major difference between the two sites is that the water table is 21 m below the ground surface at the Russell Ranch site and only 2 m below the surface at the Delta site [39]. Because of this difference, sub-irrigation in the form of irrigated ditches was used to draw up the water table and irrigate at the Delta location, whereas subsurface drip irrigation (SDI) was used at the Russell Ranch site. According to the USDA Web Soil Survey, the soil at the Russell Ranch site has an available water supply of 15.8 cm in the top 100 cm of soil, while the Delta site has 34.5 cm in the top 100 cm of soil when the soil is at field capacity [38].

In 2018 at Russell Ranch, experiments were performed on a single 1-acre plot that was conventionally managed without a cover crop. The 2018 experiment explored the effect of imposed N and water deficiency while the 2019 experiment at Russell Ranch explored the difference between a conventionally managed and an organically managed system across six different 1-acre plots. Also, the Russell Ranch 2018 experiment was on a plot that was in a maize-tomato rotation with a winter fallow, while the 2019 experiment was conducted on plots that were in a maize-tomato rotation with a winter legume cover crop. The Delta 2018 treatments were imposed across a 2-acre area in a field that was conventionally managed as a continuous maize system.

At the Russell Ranch 2018 site, hereafter referred to as the experimental site, NuTech OA713 (an F1 hybrid with 113-day maturity) maize (*Zea mays* L.) was planted on 21 April 2018 on a 1-acre plot with 5-foot beds at a rate of 79,300 seeds ha$^{-1}$, in two rows per bed. Five treatments replicated four times combined three irrigation rates (25%, 50%, and 100% ET replacement) and three nitrogen fertilizer rates (28 kg N ha$^{-1}$, 117 kg N ha$^{-1}$, 235 kg N ha$^{-1}$ of UAN-32) (Table 1 and Figure 1). A starter fertilizer (254 L ha$^{-1}$ of 8-24-6) was applied via a shank at a 10 cm depth at planting for a total of 28 kg N ha$^{-1}$ (Passport, CoreAgri, Arroyo Grande, CA, USA). An initial soil sample for N content was taken the day after planting and the starter fertilizer application. Soil nitrate-N was estimated using a nitrate quick test (Industrial Test Systems, Rock Hill, SC, USA) and an in-house calibration curve to calculate an approximate lab nitrate value [40]. All remaining N fertilizer was applied using SDI over five weekly in-season fertigation events between 5 June and 3 July 2018, corresponding to the seventh leaf (V7) and tasseling (VT) stages of crop development (Table 2). Evapotranspiration (ET) for irrigation purposes was estimated using the California Irrigation Management Information System (CIMIS) and previously determined California-based crop coefficients for maize [41,42]. Irrigation treatments were imposed by varying the number of days per week that the treatment received irrigation through SDI and were imposed from 29 May (approximately V6) to 15 July (approximately R3). The 100%, 50%, and 25% ET irrigation treatments received irrigation for six, three, and two days per week respectively. The amount of irrigation was measured using OMNI flow meters (Sensus by Xylem, Morrisville, NC, USA) at each drip line (Netafim Streamline, Fresno, CA, USA, 0.60 L/h, non-PC) for one of the four experimental replicates. An overall average flow through each dripline (103 L/h), the hours of irrigation per day, and the number of days of irrigation for each treatment was used to calculate the total irrigation for each treatment (Table 1). After 15 July, all treatments received 100% ET irrigation. Nitrogen and irrigation treatments were imposed to coincide with the period of maximum seasonal N uptake rates and to avoid effects of terminal drought stress. Each treatment was imposed on a minimum of a 5-foot bed containing two rows of maize. Treatments were randomized within each replicate over the one-acre plot with a buffer between each replicate of 8 rows of maize. Buffers received 235 kg N ha$^{-1}$ and 100% ET irrigation (Figure 1).

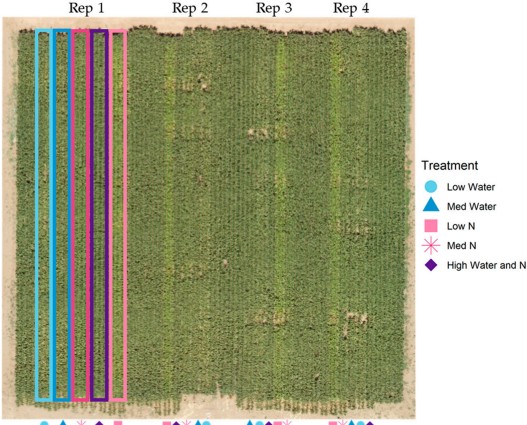

**Figure 1.** A map representing the treatment structure consisting of replicates (Rep) and water and nitrogen (N) treatments at the Russell Ranch 2018 site.

**Table 1.** A description of the applied nitrogen and irrigation treatments imposed at all sites included in this study.

| Site | Year | Treatment Description | Pre-Emergence Soil Nitrate-N 0–30 cm (kg NO$_3$-N ha$^{-1}$) [1] | Total Nitrogen Applied (kg N ha$^{-1}$) | Fertilizer Type | Irrigation Treatment (%) | Actual Irrigation Applied (mm) |
|---|---|---|---|---|---|---|---|
| Russell Ranch | 2018 | Low Water | 29 | 235 | UAN 32 | 25 | 403 |
| Russell Ranch | 2018 | Medium Water | 29 | 235 | UAN 32 | 50 | 454 |
| Russell Ranch | 2018 | Low N | 29 | 28 | UAN 32 | 100 | 615 |
| Russell Ranch | 2018 | Medium N | 29 | 117 | UAN 32 | 100 | 620 |
| Russell Ranch | 2018 | High N and Water | 29 | 235 | UAN 32 | 100 | 626 |
| Delta | 2018 | No pre-plant N | 15 | 34 | 8-24-6 Starter | 100 | — |
| Delta | 2018 | Pre-plant N applied | 15 | 148 | Anhydrous Ammonia | 100 | — |
| Russell Ranch | 2019 | Mineral N | — | 235 | UAN 32 | 100 | — |
| Russell Ranch | 2019 | Organic N | — | 269 | Composted Poultry Manure | 100 | — |

[1] One composite soil sample was used to estimate pre-emergence soil nitrate-N levels across all treatments in a single site-year.

**Table 2.** A description of maize growth stages with commonly used abbreviations.

| Stage | Description |
|---|---|
| VE | Emergence |
| V1 | First leaf collar visible |
| V2 | Second leaf collar visible |
| V(n) | nth leaf collar visible |
| VT | Tasselling—tassel has fully emerged and last branch is fully visible above leaf |
| R1 | Silking—silks are visible at the tip of the ear |
| R2 | Blister—kernels form into a white blister |
| R3 | Milk—kernels turn yellow and release milky fluid when punctured |
| R4 | Dough—kernels continue accumulating starch and change to a doughy consistency |
| R5 | Dent—all kernels are denting indicating a loss in moisture |
| R6 | Physiological Maturity—kernel black layer has formed |

At the Delta site, hereafter referred to as validation site 1, LG Seeds Maize ES7514 (an F1 hybrid with 114-day maturity) was planted at a rate of 85,900 seeds ha$^{-1}$ with a starter fertilizer of 318 L ha$^{-1}$ of an 8-24-6 liquid on 14 May 2018 made from a blend of UAN-32, phosphoric acid, zinc, and potash. A pre-plant fertilizer application of anhydrous ammonia (82% N) was applied on 10 May at a rate of 140 kg ha$^{-1}$ in 18.3-m-wide strips within the field. Select regions of the field did not receive anhydrous ammonia to create areas of low nitrogen (34 kg N ha$^{-1}$) and high nitrogen (148 kg N ha$^{-1}$) in two in-field replications (Table 1). A soil sample was taken prior the fertilizer application and was used to estimate soil nitrate-N using the same method as at the Russell Ranch 2018 site. Uniform irrigation was applied over three events in-season using sub-irrigation in the form of irrigated ditches, which raise the field water table into the root zone. Although the amount of irrigation was not measured, irrigation scheduling was done to ensure that plants did not experience water stress. Within each of the two replicated zones, two spatially distinct subsamples were taken for a total of four field samples for each treatment.

At the Russell Ranch 2019 site, hereafter referred to as validation site 2, NuTech OA713 maize was planted on 3 June 2019 on six one-acre plots with 5-foot beds at a rate of 79,300 seeds ha$^{-1}$. Seeds were

planted in two rows per bed. A composted poultry manure fertilizer was applied at a rate of 269 kg N ha$^{-1}$ on 23 October 2018 to three of the six 1-acre plots as the Organic N treatment. All treatments had a bell bean, lana vetch, and oat seed cover crop mix planted on 5 November 2018. The cover crop was mowed on 22 February 2019 and disked twice on 23 February 2019. On 23 April 2019 the other three of the six 1-acre plots received a starter of 8-24-6 for 28 kg N ha$^{-1}$ (Passport, CoreAgri, Arroyo Grande, CA, USA) as the Mineral N treatment. All plots were then irrigated via sprinklers and cultivated on 29 May 2019. Remaining fertilizer for the Mineral N treatment was applied using SDI over four weekly in-season fertigation events between 18 July and 8 August 2019 (Table 1). Each treatment received greater than 100% ET irrigation through SDI to account for soil moisture depletion by the winter cover crops.

### 2.2. Changes in N Uptake over Time

In-season biomass harvests were used to estimate biomass production and nitrogen uptake. For the experimental site, six in-season biomass harvests were completed at the V6, V9, V14, blister (R2), dent (R5), and full maturity (R6) growth stages. For a description of maize growths stages, see Table 2. The growth stages of V6, V14, and R2 were selected for further analysis for the following reasons: V6 is an early vegetative stage and was the time point when treatments were imposed at the experimental site; V14 is a late vegetative stage during the period of rapid growth and N uptake in maize; and R2 represents the period of maximum stress (irrigation treatments were terminated after R2 and, subsequently, all treatments received 100% ET irrigation). The growth stages at harvest were V14, V18, and R6 for validation site 1 and V14, R2, and R6 for validation site 2. For each hand harvest, sixteen plants were cut from each experimental unit to obtain an aboveground biomass estimate. A subsample of those same plants was then used to determine plant moisture and nitrogen concentration. At the R2 and later growth stages, cobs and grain were separated from the biomass sample at harvest. For the experimental site and validation site 1, grain yield at the plot scale was estimated via hand harvest and adjusted post-hoc by an increase of 8.5% to coincide with yield data from the field-scale combine harvest estimate of the same treatment. The grain yields for validation site 2 were directly calculated via a field-scale combine harvest. All plant subsamples were dried in an oven at 60 degrees C until they reached a stable weight and then ground to 4 mm using a Wiley Mill Grinder (Model 4, Thomas Scientific, Philadelphia, PA, USA) for non-grain biomass and a coffee grinder for grain. The samples were further ground into a fine powder using a ball mill before being analyzed with a near-infrared spectrometer for crude protein and nitrogen content using an unfermented maize silage calibration equation (Foss Model 6500, Foss North America, Eden Prairie, MN, USA). A subsample of biomass and grain samples were likewise analyzed for total N concentration using an elemental analyzer with a continuous flow isotope ratio mass spectrometer (IRMS) (Sercon Ltd., Cheshire, UK). These values were used to generate a calibration curve for the NIR method and estimate absolute N uptake values. The slope of the relationship between the NIR method values and the elemental analyzer method values was used to adjust the absolute N concentration across all samples.

Growing degree days (GDD) were calculated using the daily maximum and minimum air temperatures measured using a NOAA reference climatological station operated by the Department of Land, Air, and Water Resources at University of California, Davis [43]. The single sine method of calculation was used with an effective maximum temperature of 30 °C and an effective minimum effective temperature of 10 °C. A quantitative relationship between changes in total N uptake and GDD was generated at the experimental site using the lm() function in R 3.6.1 to quantify a second-degree polynomial model with GDD and an interaction with treatment [44]:

$$\text{Total N Uptake}_i = \beta_0 + \beta_1 \text{GDD} + \beta_2 \text{GDD}^2 + \beta_3 \text{Treatment} +$$
$$\beta_4 \text{GDD*Treatment} + \beta_5 \text{GDD}^2\text{*Treatment} + \varepsilon_i. \tag{1}$$

R squared and *p*-values were determined using a type III ANOVA from the car() package (3.0-3) [45].

## 2.3. Calculation of Vegetation Indices

Multispectral images were recorded at each site on a weekly basis using an unmanned aerial vehicle (UAV). A DJI Matrice 100 (DJI, Shenzhen, China) equipped with a DJI Zenmuse Z3 RGB camera and a MicaSense RedEdge-M multi-spectral camera (MicaSense, Seattle, WA, USA) flew 60 m above the ground in a north-to-south direction with an 85% overlap of images. The center wavelengths of each band captured with the RedEdge-M camera are blue (475 nm), green (560 nm), red (668 nm), red edge (717 nm), and near infrared (840 nm). Flights were completed within one hour of solar noon on clear days to avoid cloud and shadow interference in images. An image was also taken of a calibration panel of known reflectance in each of the bands of the RedEdge-M camera. After image capture, images were stitched and calibrated using Pix4Dmapper processing software (1.3.0 and 1.4.0) (Pix4D S.A., Lausanne, Switzerland) to a resolution of 5 cm per pixel.

Average pixel values were extracted within each experimental unit using QGIS version 3.10, excluding areas that had been harvested or were impacted by edge effects [46]. The average values were then used to calculate the Normalized Difference Red Edge (NDRE) and Green Leaf Index (GLI) based on previously developed indices [30,47]:

$$NDRE = \frac{NIR - Red\ Edge}{NIR + Red\ Edge} \tag{2}$$

$$GLI = \frac{2Green - Red - Blue}{2Green + Red + Blue} \tag{3}$$

These indices were then used to generate normalized indices using the following equations for the normalized NDRE (NDRE$_N$) and the normalized GLI (GLI$_N$):

$$NDRE_N = \frac{NDRE_m}{NDRE_{95}} \tag{4}$$

$$GLI_N = \left(\frac{1}{GLI_m}\right) \Big/ \left(\frac{1}{GLI_m}\right)_{95} \tag{5}$$

where the subscript m indicates the indices calculated using Equations (2) and (3) based on average measured reflectance values within an experimental unit, and the subscript 95 indicates the 95th percentile value for the average measured index across the experimental units for each flight. Similarly, the blue reflectance was normalized into a Blue Reflectance Index (BRI$_N$) using

$$BRI_N = \left(\frac{1}{Blue_m}\right) \Big/ \left(\frac{1}{Blue_m}\right)_{95} \tag{6}$$

This ensured that variation between flights was normalized and that, for each index, high relative values corresponded to the healthiest plants. Of note: since the 95th percentile (not the maximum value) was used in the calculation, normalized index values can be greater than 1.

## 2.4. Selection of Sensitive Indices

One way linear ANOVAs were used to analyze the relationship between the treatment structure and cob N uptake or total aboveground biomass at the R2 growth stage for the experimental site using the equation

$$Cob\ N\ Uptake_i\ OR\ Total\ Biomass_i = \beta_0 + \beta_1 Treatment + \varepsilon_i. \tag{7}$$

To identify indices most sensitive to water and nitrogen stress separately, data was split into two groups based on the type of deficiency imposed with the high water & N treatment in both groups. An ANOVA analysis of the mixed linear model was completed using of the format

$$\text{Cob N Uptake}_i \text{ OR Total Biomass}_i = \beta_0 + \beta_1 \text{Index} + \varepsilon_i, \text{random} = 1|\text{Replicate}. \qquad (8)$$

where Index was either $\text{NDRE}_N$, $\text{GLI}_N$, or $\text{BRI}_N$. Marginal r squared values were calculated for each model using the r.squaredGLMM() function from the MuMIn() package (1.43.10) [48] and *p*-values were found using the anova.lme() function from the nlme() package (3.1-140) [49].

## 2.5. Development of the Combined Index

Reflectance-based groups for the $\text{NDRE}_N$, $\text{GLI}_N$, or $\text{BRI}_N$ indices were classified as "high" and "low" for each index using Gaussian mixture modeling, a form of unsupervised univariate classification, via the normalmixEM() function from the mixtools() package (1.1.0) in R version 3.6.1 [50] (Figure A1). The average index values for each experimental unit were used to run 1000 iterations of the normalmixEM() function with a set value of two possible populations as the output. The most often occurring convergence of the 1000 iterations were used to initialize a final fit. Outputs of the function include means and standard deviations of the data points in the mixture modeling output populations. These statistical outputs were used to develop separate thresholds and groupings for each site-year and stage of growth such that the data in each group was as evenly distributed as possible (Figure A1). These quantitative thresholds were used to place each experimental unit for each site-year and growth stage into "high' or "low" groups based on each index value. The diagram in Figure 2 illustrates how the resulting groups were used to split the experimental units into four different groups hereafter referred to as the Combined Index (CI).

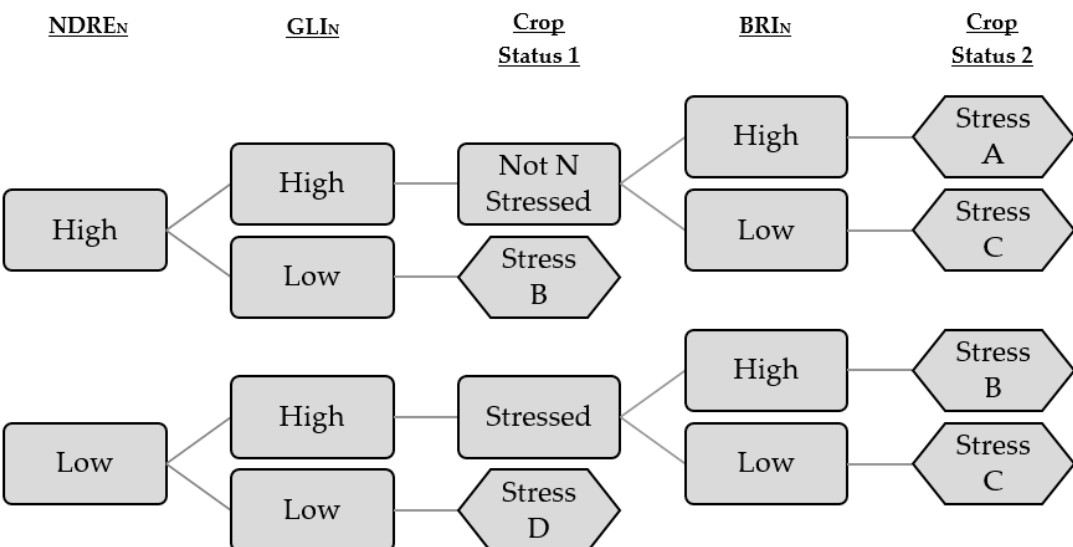

**Figure 2.** Combined Index (CI) used to separate experimental units into one of four nitrogen (N) or water-based crop status outcomes (Stress A, B, C, and D) based on combined normalized green leaf index ($\text{GLI}_N$), normalized difference red-edge ($\text{NDRE}_N$), and blue reflectance index ($\text{BRI}_N$) measurements.

## 2.6. Assessment of the Combined Idex at the Experimental Site

The effectiveness of the CI at separating the experimental units into distinct groups was assessed by comparing the mixed linear relationships between biophysical outcomes to the experimental treatments, CI-based categories generated at R2 and V14, $\text{NDRE}_N$ at R2, and $\text{NDRE}_N$ at V14. The biophysical outcomes used to assess these different indices were total N uptake at V14, total N uptake at R2, cob yield at R2, R5 biomass yield, and grain yield. Replicate was included as a random factor in

all models. Marginal r squared values were calculated for each model using the r.squaredGLMM() function from the MuMIn() package (1.43.10) [48] and *p*-values were found using the anova.lme() function from the nlme() package (3.1-140) [49]. To compare the effectiveness of the CI and the $NDRE_N$ alone at explaining variation in the biophysical outcomes, a Davidson-MacKinnon J test was performed on the non-nested linear models using the function jtest() from the lmtest() package (0.9-37) [51,52].

*2.7. Assessment of the Combined Index at the Validation Sites*

After the CI was derived for each site-year, the contribution of each individual index ($NDRE_N$, $GLI_N$, and $BRI_N$) and the CI to the relationship with the biophysical outcomes was assessed using mixed linear models for the experimental site and the combined validation sites using the lme() and anova.lme() functions from the nlme() package (3.1-140) as follows [49]:

$$\text{Grain Yield}_i \text{ OR Grain N Uptake}_i = \beta_0 + \beta_1 \text{Index} + \varepsilon_i, \text{random} = 1|\text{Replicate} \tag{9}$$

for the experimental site and

$$\text{Grain Yield}_i \text{ OR Grain N Uptake}_i = \beta_0 + \beta_1 \text{Index} + \varepsilon_i, \text{random} = 1|\text{Site-Year}|\text{Management}|\text{Replicate} \tag{10}$$

for the validation sites where Index is CI, $NDRE_N$, $GLI_N$, or $BRI_N$. For the validation sites, site-year was either Delta 2018 or Russell Ranch 2019 and management was differentiated between Conventional and Organic at RR 2019 only.

Finally, to illustrate the consistency or inconsistency of individual indices across different sites and stages of growth, individual pixel values were extracted from the same area used to calculate the mean index using the extract() function from the raster() package (3.0-7) [53]. Index values were normalized by dividing the raw values by the 95th percentile of the highest reflectance values for each image individually and plotted as histograms using the ggplot() function and geom_histogram() from the ggplot2() package (3.2.1) [54]. All statistical analyses were completed in R version 3.6.1 [44].

# 3. Results

*3.1. Changes in N Uptake and NDRE over Time*

The N and water gradients imposed at the experimental site led to differentiation in plant productivity in terms of aboveground biomass and total N uptake. There was a large difference in total N uptake between the non-stressed treatment (sufficient N & water) and the low N treatment (Figure 3). NDRE values began to indicate differences between the two treatment extremes at approximately the same time as those differences were manifesting in terms of total N uptake by the plants. As the season progressed, the difference between these two treatments became more pronounced. For example, at V14, a relatively late vegetative growth stage, the total N uptake for the low N treatment was only 75% of the value of the high N & water treatment. The difference in total N uptake between these two treatments increased by 87% from V14 to R2. Similarly, the difference in measured NDRE between these two treatments increased by 132% from V14 to R2 (Figure 3). The total N uptake and NDRE values measured for all other treatments fell within these two extremes over the course of the season, suggesting that changes in NDRE over the course of crop development corresponded to changes in N uptake.

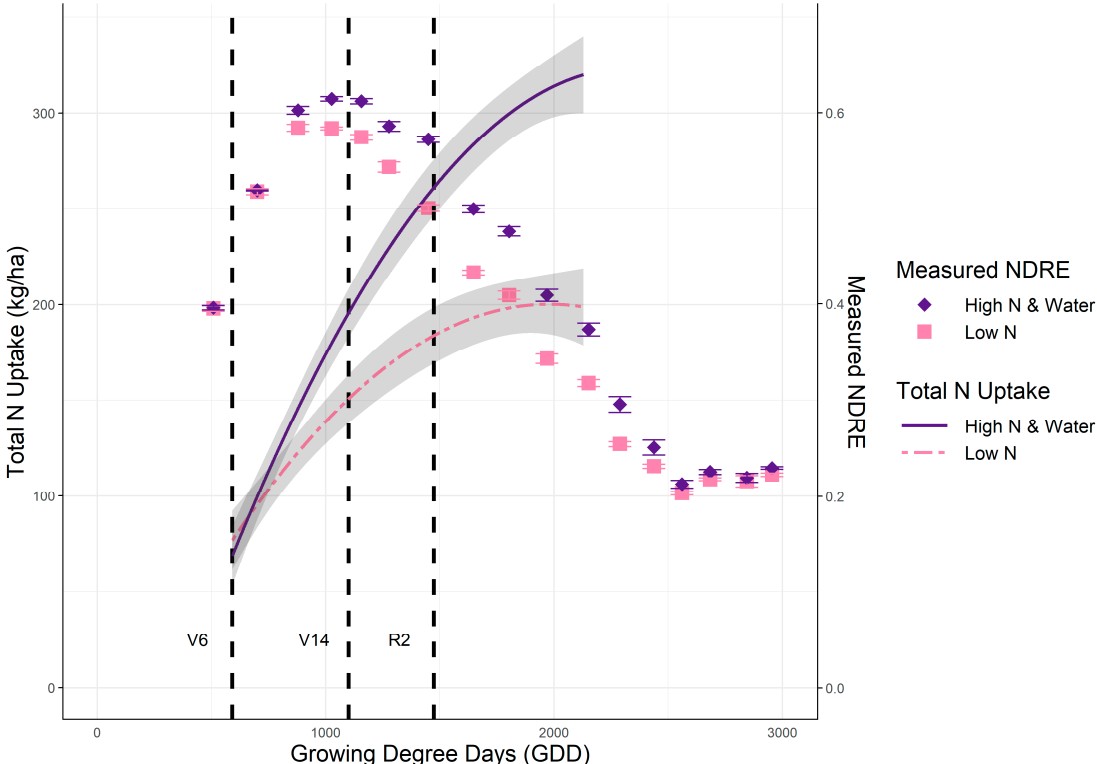

**Figure 3.** Changes in normalized difference red-edge (NDRE) reflectance and total nitrogen (N) uptake across the season (measured in GDD) for the low nitrogen treatment and the unstressed treatment at the experimental site (Russell Ranch 2018). Total N uptake curves are shown with confidence intervals ($\alpha = 0.05$) and NDRE measurements are shown with standard error bars. Vertical dashed black lines indicate the timing of growth stages (V6, V14, and R2) at the experimental site.

### 3.2. Selection of Sensitive Indices

The linear relationships between normalized reflectance values and cob N uptake or total biomass at R2 were used to determine the relative sensitivity of the indices to N and water stress at the experimental site (Russell Ranch 2018). Cob N uptake at R2 was selected to identify nitrogen stress because it had a strong linear relationship with the N treatments of the study ($r^2 = 0.91$, $p < 0.001$). Total biomass at R2 was selected to identify water stress since it had a strong linear relationship with the irrigation treatment structure ($r^2 = 0.84$, $p < 0.005$).

NDRE$_N$ had a strongly linear relationship with biophysical outcomes at R2 regardless of the type of deficiency imposed (Table 3). NDRE$_N$ had a linear relationship with cob N uptake ($r^2 = 0.89$, $p < 0.001$) and total biomass at R2 ($r^2 = 0.40$, $p < 0.05$) for the nitrogen treatments and the water treatments ($r^2 = 0.67$, $p < 0.005$ and $r^2 = 0.49$, $p < 0.01$ respectively). GLI$_N$ had a stronger linear relationship with cob N uptake and total biomass at R2 for the nitrogen treatments ($r^2 = 0.86$, $p < 0.001$ and $r^2 = 0.45$, $p < 0.05$ respectively) than for the water treatments ($r^2 = 0.07$, $p = 0.38$ and $r^2 = 0.32$, $p < 0.01$ respectively) (Table 3). All the water stressed treatments had a GLI$_N$ at R2 greater than 0.95 while the medium N and low N treatments had GLI$_N$ values less than 0.95 at R2, which indicates that GLI$_N$ may be selectively sensitive to nitrogen stress versus water stress (Table 4). In contrast, the BRI$_N$ was more sensitive to the total biomass of the water treatments ($r^2 = 0.41$, $p < 0.01$) compared to the nitrogen treatments ($r^2 = 0.30$, $p < 0.05$), indicating that it may be more sensitive to water stress than the GLI$_N$ (Table 3).

**Table 3.** Anova results relating the normalized multi-spectral indices green leaf index ($GLI_N$), blue reflectance index ($BRI_N$) and normalized difference red-edge ($NDRE_N$) to total biomass production or cob N uptake for the experimental site at R2 using Equation (8). NS = not significant at $\alpha = 0.05$, $r^2$ = coefficient of determination.

| Treatment Group | Measurement | Index | $r^2$ | *p*-Value |
|---|---|---|---|---|
| Nitrogen | Cob N Uptake | $GLI_N$ | 0.86 | <0.001 |
| Nitrogen | Total Biomass | $GLI_N$ | 0.45 | <0.05 |
| Water | Cob N Uptake | $GLI_N$ | 0.07 | NS |
| Water | Total Biomass | $GLI_N$ | 0.32 | <0.1 |
| Nitrogen | Cob N Uptake | $BRI_N$ | 0.77 | <0.001 |
| Nitrogen | Total Biomass | $BRI_N$ | 0.30 | <0.05 |
| Water | Cob N Uptake | $BRI_N$ | 0.54 | <0.01 |
| Water | Total Biomass | $BRI_N$ | 0.41 | <0.01 |
| Nitrogen | Cob N Uptake | $NDRE_N$ | 0.89 | <0.001 |
| Nitrogen | Total Biomass | $NDRE_N$ | 0.40 | <0.05 |
| Water | Cob N Uptake | $NDRE_N$ | 0.67 | <0.005 |
| Water | Total Biomass | $NDRE_N$ | 0.49 | <0.01 |

**Table 4.** Thresholds for the normalized blue reflectance index ($BRI_N$), green leaf index ($GLI_N$), and normalized difference red-edge ($NDRE_N$) values generated using univariate Gaussian mixture modeling. Thresholds were used to indicate "high" and "low" relative groupings for each index at either the 14th leaf stage (V14) or blister (R2) stage of growth.

| Site and Growth Stage | Thresholds | | |
|---|---|---|---|
| | $BRI_N$ | $GLI_N$ | $NDRE_N$ |
| Experimental Site R2 | 0.95 | 0.95 | 0.93 |
| Experimental Site V14 | 0.96 | 0.93 | 0.98 |

Based on these relationships, the combined index (CI) was generated using $BRI_N$, $GLI_N$, and $NDRE_N$ in tandem to separate water stress from N stress as follows. The stage-specific thresholds for $NDRE_N$, $GLI_N$, and $BRI_N$ derived from Gaussian mixture modeling were used to categorize each experimental unit into either "high" or "low" reflectance groups for the three indices at V14 and R2 (Table 4, Figure A1a–f). The CI rubric (Figure 2) was then used in combination with the reflectance groupings to separate the experimental units into the one of the four stress classification groups. Based on the relationships presented in Table 3, $NDRE_N$ was used to classify initial stress level. Next, the $GLI_N$ was used to magnify the sensitivity to nitrogen stress measured via the $NDRE_N$. Subsequently, the $BRI_N$ was used to magnify the sensitivity to the non-N form of stress present. As such, the Crop Status categories of A, B, C, and D in Figure 2 are interpreted as follows: non-stressed, medium N stressed, water-stressed, high N stressed. Experimental units that had high values for all indices were non-stressed while experimental units that were categorized as low in either $NDRE_N$ or $GLI_N$ were categorized as medium N stressed. Where it was determined that N stress was not severe and the $BRI_N$ was low, experimental units were categorized as water-stressed.

*3.3. The Combined Index at the Experimental Site*

Categorical responses in crop productivity were analyzed both as a function of the experimental treatment structure and as a function of the groupings created via the CI at R2 (Table 5). The low N treatment had the greatest effect on yields with the greatest yield penalty for final grain ($-7195$ kg ha$^{-1}$) and R5 biomass ($-4915$ kg ha$^{-1}$) (Table 5). Water stress had less impact on final grain yields ($-5242$ kg ha$^{-1}$ for the low water treatment and $-3046$ kg ha$^{-1}$ for the medium water treatment). R5 biomass yield was similarly affected by water stress and nitrogen stress with yield penalties of $-4915$ kg ha$^{-1}$ for the low water treatment and $-3731$ kg ha$^{-1}$ for the medium water treatment compared to yield penalties of $-4961$ kg ha$^{-1}$ and $-2483$ kg ha$^{-1}$ for the low N and medium N

treatments respectively (Table 5). Overall, the imposed treatments explained 82% of the variation in final grain yield ($p < 0.001$) and 76% of variation in R5 biomass yield ($p < 0.001$) (Table 6).

**Table 5.** A summary of the grain and dent (R5) biomass yields for the imposed water or nitrogen (N) treatments and Combined Index generated at blister (R2) at the experimental site. SE = standard error, CV = coefficient of variation.

| Method | Group | Grain Yield (15.5% Moisture) | | | | R5 Biomass Yield | | | |
|---|---|---|---|---|---|---|---|---|---|
| | | Yield (kg ha$^{-1}$) | SE (kg ha$^{-1}$) | n | CV (%) | Yield (kg ha$^{-1}$) | SE (kg ha$^{-1}$) | n | CV (%) |
| Treatment Structure | Low Water | 12,089 | 1207 | 4 | 20 | 19,310 | 683 | 4 | 7 |
| | Medium Water | 14,285 | 1089 | 4 | 15 | 20,494 | 669 | 4 | 7 |
| | Low N | 10,136 | 511 | 4 | 10 | 19,264 | 491 | 4 | 5 |
| | Medium N | 13,181 | 399 | 4 | 6 | 21,742 | 519 | 4 | 5 |
| | High Water & N | 17,331 | 713 | 4 | 8 | 24,225 | 952 | 4 | 8 |
| R2 Combined Index | Water Stressed | 13,038 | 1141 | 6 | 21 | 20,089 | 656 | 6 | 8 |
| | High N Stress | 10,136 | 511 | 4 | 10 | 19,264 | 491 | 4 | 5 |
| | Medium N Stress | 13,181 | 399 | 4 | 6 | 21,742 | 519 | 4 | 5 |
| | Non-stressed | 16,099 | 933 | 6 | 14 | 22,596 | 1193 | 6 | 13 |

**Table 6.** Linear relationships of the Combined Index (CI) groupings, relative normalized difference red edge (NDRE$_N$) alone, and the treatment structure to the 14th leaf stage (V14) total nitrogen (N) uptake, blister (R2) total N uptake, R2 cob yield, dent (R5) biomass yield, and final grain yield for the experimental site. $r^2$ = coefficient of determination.

| Growth Stage | Index | V14 Total N Uptake | | R2 Total N Uptake | | R2 Cob Yield | | R5 Biomass Yield | | Grain Yield | |
|---|---|---|---|---|---|---|---|---|---|---|---|
| | | $r^2$ | $p$-Value | $r^2$ | $p$-Value | $r^2$ | $p$-Value | $r^2$ | $p$-Value | $r^2$ | $p$-Value |
| V14 | CI | 0.14 | 0.39 | 0.35 | <0.05 | 0.23 | 0.12 | 0.10 | 0.57 | 0.27 | 0.14 |
| | NDRE$_N$ | 0.10 | 0.13 | 0.49 | <0.001 | 0.53 | <0.001 | 0.17 | <0.05 | 0.45 | <0.01 |
| R2 | CI | 0.30 | <0.05 | 0.62 | <0.001 | 0.67 | <0.001 | 0.35 | <0.005 | 0.53 | <0.005 |
| | NDRE$_N$ | 0.16 | <0.05 | 0.68 | <0.001 | 0.76 | <0.001 | 0.30 | <0.005 | 0.50 | <0.001 |
| Treatment Structure | | 0.28 | 0.08 | 0.72 | <0.001 | 0.80 | <0.001 | 0.66 | <0.001 | 0.68 | <0.001 |

The CI groupings successfully placed all low N plots into the High N Stress category, all medium N plots into the Medium N Stress category, and all high N plots into the Non-stressed category (Table 5). The CI permitted three categories relating to nitrogen stress (Non-stressed, Medium N Stress, and High N Stress) while only allowing two different categories relating to water stress (Non-stressed, and Water Stressed) (Figure 2). Thus, the CI identified 100% of the nitrogen deficient experimental units both in terms of the stress-type and stress-severity. The CI also successfully identified 75% of the water deficient experimental units as being water stressed and placed two medium water experimental units into the Non-stressed group. The total success rate of the CI across the entire treatment structure was 90%, with 2 of the 20 total plots misidentified as being-non-stressed. In addition, the CI groups delineated similar effects of stress on yields as the treatment structure. The high N stress group had the greatest grain yield penalty ($-5963$ kg ha$^{-1}$) as well as the greatest R5 biomass yield penalty ($-3332$ kg ha$^{-1}$) (Table 5). Compared to the high N stress group, CI-identified water stress had less effect on grain yields with a yield penalty of $-3061$ kg ha$^{-1}$. R5 biomass yields for the CI-delineated Water Stressed group were impacted similarly to the high N stress group ($-2507$ kg ha$^{-1}$) (Table 5).

Although the CI successfully differentiated between nitrogen and water stress at R2, the CI generated at V14 using the same methods was less effective at identifying the type and severity of stress. At V14, the CI successfully identified 100% of the nitrogen stressed experimental units but only 63% were assigned the correct severity of high or medium N stress. The CI only correctly identified 38% of the water stressed experimental units, assigning three of the low water and two of the medium water experimental units to the Non-stressed designation. Overall, the CI at V14 successfully identified 60% of experimental treatments to the correct stress type and severity of stress. Compared to the CI at V14, NDRE$_N$ measured at V14 explained more variation in the total N uptake at

R2 ($p < 0.05$) and the cob yield at R2, ($p < 0.001$) according to non-nested model comparison (Table 6). In addition, $NDRE_N$ measured at R2 explained more variation in and cob yield ($p < 0.05$) than the CI at R2 (Table 6). Taken together, these results indicate that the $NDRE_N$ was generally more effective than the CI at identifying gradients of stress, but the CI may have a relatively narrow ability to differentiate categorically between N and water stress.

*3.4. The Combined Index at the Validation Sites*

To further quantify and compare the efficacy of the CI and $NDRE_N$ at identifying degrees and types of crop stress, the same methods were applied at two independent site-years where gradients of N availability were present, but where no water stress had been intentionally imposed (see Materials and Methods Section 2.1). At validation site 1, the CI was generated from reflectance values recorded at the V18 stage of growth, with resulting gaussian mixture model threshold values of 0.98, 0.99, and 0.97 for $BRI_N$, $GLI_N$, and $NDRE_N$ respectively. The CI at validation site 1 successfully identified 100% of the low N experimental units as under high N stress and 75% of the high N experimental units as being non-stressed. However, the CI misidentified the high N experimental unit with the greatest grain yield at the site as being water stressed.

At validation site 2, the CI was generated at R2 with gaussian mixture modeling threshold values of 0.98, 0.98, and 0.93 for $BRI_N$, $GLI_N$, and $NDRE_N$ respectively. It identified all the mineral N experimental units as being in the medium nitrogen stress group and all the organic N experimental units in the water-stressed group. Because the mineral N treatment received sufficient N and both treatments received daily irrigation (at similar rates as the non-stressed treatment at the experimental site (Table 1)) and did not exhibit any other typical symptoms of water stress like leaf rolling or stunted growth, this was likely a false indication of relative stress.

To investigate why the CI was less effective at the validation sites at predicting grain yield ($r^2 = 0.41$, $p = 0.31$) and grain N uptake ($r^2 = 0.61$, $p = 0.18$) (Table 7), the linear relationships between $NDRE_N$, $GLI_N$, or $BRI_N$ and grain yield or N uptake for the experimental site as well as the validation sites were examined. In comparing the marginal r squared values for $NDRE_N$ with both grain yield and grain N uptake between the experimental site and the validation sites, the values were very similar (Table 7), indicating that $NDRE_N$ was sensitive to differences in yield regardless of the type of stress. The $GLI_N$ did not significantly explain variation in grain yield or grain N uptake at the validation sites ($r^2 = 0.14$ for grain yield and $r^2 = 0.24$ for grain N uptake) although it had at the experimental site ($r^2 = 0.38$ for grain yield and $r^2 = 0.60$ for grain N uptake) (Table 7). The $BRI_N$ correlated to both grain yield ($r^2 = 0.39$) and grain N uptake ($r^2 = 0.36$) for the experimental site, and had an equivalent or stronger linear relationship to the grain yield ($r^2 = 0.48$) and grain N uptake ($r^2 = 0.45$) for the validation sites (Table 7). Because the difference between the experimental sites and validation sites was that no water stress was intentionally imposed at the validation sites, this result indicates that the $GLI_N$ (and by extension the CI) may not be useful unless water deficiency exists.

Relative individual pixel reflectance values were extracted to explore the difference in results between sites and investigate whether the timing of measurement influenced the CI efficacy. At both a late-vegetative (V18) and an early-reproductive (R2) stage of growth, the site-based differences in blue pixel distribution were small. Standard deviations of the blue pixel values for validation site 1 and the experimental site at V18 were 0.102 and 0.110, respectively, and changed to 0.098 and 0.109, respectively, by R2 (Figure 4). Despite the broad similarities, the overall variability in blue pixels at the experimental site was larger. At V18 the GLI pixels mean was the same at the experimental site and validation site 1 with a similar standard deviation (0.126 at validation site 1 and 0.116 at the experimental site) (Figure 4). However, at R2 differences in the GLI pixel distributions emerged between the two sites. The experimental site had a higher standard deviation (0.126) compared to validation site 1 (0.069), and GLI pixels at validation site 1 split into two modes, indicating a divergence in reflectance between the high and low N treatments, but a narrower and less continuous stress profile (Figure 4). This result illustrates the importance of measurement timing to the differences detected by

GLI, which only differentiated stress during the reproductive stages of growth. Also, the yield penalty at validation site 1 for the low N treatment was only −3700 kg ha⁻¹ of grain compared to −7200 kg ha⁻¹ of grain for the experimental site, which may help to explain the narrower range of GLI values at validation site 1. The standard deviation of NDRE pixels at V18 was higher at validation site 1 (0.106) compared to the experimental site (0.064), which likely reflects greater spatial variability at that site than the experimental site. The relationship reversed at R2 with standard deviations of 0.045 for the validation site 1 and 0.087 for the experimental site (Figure 4). Similar to the GLI at R2, NDRE values at R2 had a narrower range at the validation site (where the yield differences were smaller) compared to the experimental site.

**Table 7.** Compares the linear relationships between the vegetation indices normalized green leaf index (GLI$_N$), blue reflectance index (BRI$_N$), and normalized difference red-edge (NDRE$_N$) and resulting combined index (CI) measured at blister (R2) and the grain yield or nitrogen uptake at the experimental site using Equation (9) and the validation sites using Equation (10). $r^2$ = coefficient of determination

| | **Grain Yield** | | | |
| | **Experimental Site** | | **Validation Sites** | |
| **Index** | $r^2$ | $p$-Value for Index | $r^2$ | $p$-Value for Index |
|---|---|---|---|---|
| NDRE$_N$ | 0.50 | <0.001 | 0.63 | <0.05 |
| GLI$_N$ | 0.38 | <0.005 | 0.14 | 0.29 |
| BRI$_N$ | 0.39 | <0.005 | 0.48 | <0.05 |
| CI | 0.53 | <0.005 | 0.41 | 0.31 |
| | **Grain N Uptake** | | | |
| | **Experimental Site** | | **Validation Sites** | |
| **Index** | $r^2$ | $p$-Value for Index | $r^2$ | $p$-Value for Index |
| NDRE$_N$ | 0.71 | <0.001 | 0.64 | <0.05 |
| GLI$_N$ | 0.60 | <0.001 | 0.24 | 0.09 |
| BRI$_N$ | 0.36 | <0.01 | 0.45 | <0.05 |
| CI | 0.76 | <0.001 | 0.61 | 0.18 |

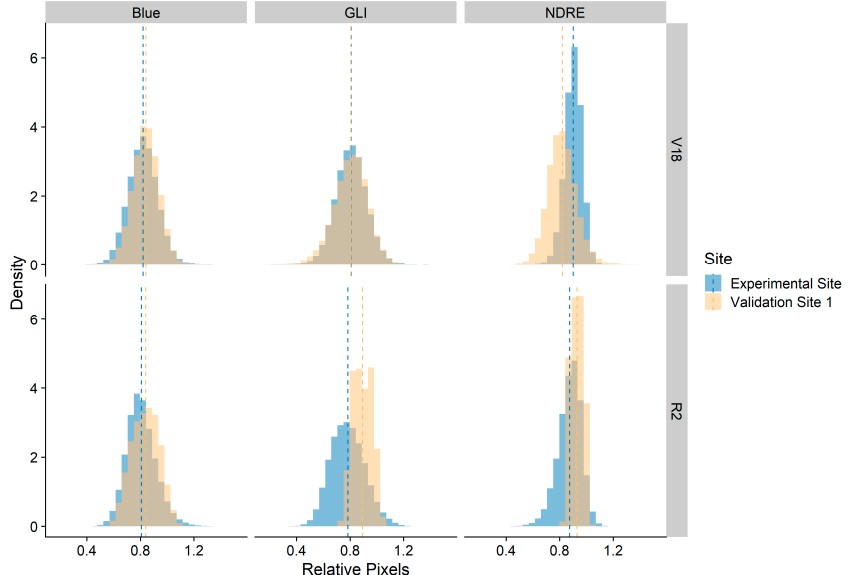

**Figure 4.** The distribution of relative values for individual pixels of blue, green leaf index (GLI), and the normalized difference red-edge (NDRE) throughout the experimental areas at the experimental site and validation site 1 at both the 18th leaf stage (V18) and blister (R2) growth stages. Individual pixel values are relative to the 95th percentile of the measured index values. Dashed lines show the mean of the pixel values for each site.

## 4. Discussion

The CI including $NDRE_N$, $GLI_N$, and $BLI_N$ was able to differentiate between N and water stress and indicate two levels of N stress severity (Table 5). By combining individual indices, the CI successfully identified 100% of the nitrogen deficient treatments and 75% of the water deficient treatments during the blister (R2) stage of growth at the experimental site. It is important to emphasize that this information was developed independent of the treatment structure based on real-time information collected at R2 (Figure 2 and Table 4), yet it delivered similar categorical information about the crop productivity outcomes as the categorical treatment structure (Table 5). Also, the CI was developed using indices that were calculated and made relative to the reflectance of the non-stressed zone in the field. Even still, the water stress designation from the CI was less accurate than the N stress designation, which may have been partially due to the greater overall variability associated with the water stress gradient relative to the N stress gradient (Table 5). In addition, in contrast to N stress, the CI provided no information about the degree of water stress.

Although the CI was successful at identifying stress at the reproductive stage, it was important to investigate the CI at an earlier growth stage when management decisions may have more of an impact on yields. V14 is a late vegetative growth stage before tasseling and ear formation, which are both important physiological stages for avoiding stress [55]. Despite its value at R2, the CI did not effectively differentiate crop stress at V14 (Table 6) even though the treatments had started to diverge in terms of plant N uptake and $NDRE_N$ (Figure 3). Nitrogen stress in maize has been detected using spectral reflectance as early as the V8 growth stage, while water stress is not as easily detected until the R1 growth stage [24]. One explanation for this insensitivity at V14 is that plant water demand had not yet reached a level that substantially exceeded soil supply in the low water treatments, thereby reducing the degree of water stress manifest and detectable at V14. In terms of nitrogen stress, fertigation events continued beyond the V14 growth stage. Since plants were being given nitrogen on a weekly basis, stress may not have manifest as strongly until after fertigation ended at tasseling. Also, since it has been shown that growth at or after the tasseling growth stage is greatly affected by water availability, the deficit at tasseling in the treatment structure could have caused greater differentiation between treatments, as detected by R2 [55]. Further, because of the dependence of the CI on the GLI (Figure 2) and the fact that the GLI did not detect differences in stress until the reproductive growth stage (Figure 4), V14 may have been too early in the crop growth cycle for the CI to be effective, even if water stress had already begun affecting crop vigor. GLI has been shown to be highly sensitive to leaf chlorophyll content but was also shown to be affected by growth stage, particularly tasseling, by Hunt et al. [20]. Also, because leaf chlorophyll content drastically drops at senescence, any vegetation index sensitive to chlorophyll would be sensitive to divergence in senescence timing [56].

The dependence of the CI on GLI was further evidenced at the validation sites, where there was significant variation between each site-year in terms of soil type, management, and environment. The GLI explained little of the variation in grain yields and grain N uptake at the validation sites compared to the experimental site (Table 7). This contrasts with the NDRE and BLI, which explained significant portions of the variability at both the experimental site and the validation sites (Table 7). At the experimental site both water and N deficiencies were imposed with temporal precision and at early stages of crop growth by treatment gradients administered via sub-surface drip irrigation. In contrast, although crops experienced N deficiency at the validation sites, no irrigation deficits were imposed, and N deficiency was imposed in two extremes, rather than as a gradient. In addition, there was greater availability of soil organic N at these locations because of the high soil organic matter at validation site 1 and the pre-season application of composted poultry manure at validation site 2. As a result, crop N deficiency likely accrued more gradually at these sites as opposed to the sudden deficiency experienced at the experimental site, which manifested in the form of more rapid rates of crop development and earlier senescence by the N-stressed treatments.

GLI has been used previously to estimate plant chlorophyll content [20], and the selective sensitivity of this index to N stress versus water stress may have been the result of more rapid

senescence of the nitrogen-deficient treatments detectable by the R2 growth stage at the experimental site (Table 3). This is further supported by the change in distribution of GLI pixels from V18 to R2 at both the experimental site and validation site 1 (Figure 4), which indicate increasing GLI differentiation at the latter stage of growth. The GLI pixel distribution suggested that there may be slightly more stress at the experimental site, as indicated by more GLI pixel values with high relative reflectance compared to the rest of the field at V18 and R2. The GLI distribution was essentially the same between the two sites at V14 but after senescence started at R2, the spread of pixels at the experimental site increased and diverged into two different modes at validation site 1. This indicates the importance of timing in interpreting reflectance measurement, particularly for GLI since it is sensitive to crop senescence.

As for the difference in pixel distribution between the two sites, this could be explained by the fact that the yield penalty at validation site 1 for the low N treatment was only $-3700$ kg ha$^{-1}$ of grain compared to $-7200$ kg ha$^{-1}$ of grain for the experimental site. There was less variation in yields at validation site 1 so less variation in reflectance between the two treatments would be expected. Therefore, unless and until nitrogen deficiency results in differentiation in rates of senescence, GLI may not provide unique information separate from that provided by the NDRE and BLI. Indeed, when GLI was removed from the CI at validation site 1 the CI with only NDRE$_N$ and BRI$_N$ reached the same conclusion as the full CI. NDRE$_N$ alone successfully identified 100% of the low N experimental units in the "low" NDRE$_N$ group and 100% of the high N experimental units in the "high" NDRE$_N$ group. At validation site 2, if the GLI$_N$ was omitted, the CI correctly identified all mineral N treatment experimental units as being non-stressed and all the organic N treatment experimental units as being under high stress, potentially co-limited by N and water. Since the validation sites did not experience directly imposed water stress, the ability of the GLI to identify N stress using differing senescence patterns was not useful. Since neither validation site included a water stressed treatment, testing the efficacy of the CI at an additional site-year with water deficiency would be a valuable next step. It would also be useful to test the CI in an environment that combines water and N stress to understand how effectively the CI can differentiate stress when the types of stress are combined.

Compared to the methods presented here, a more complicated decision tree analysis was used by Peña-Barragán et al. to differentiate between thirteen different crops over a large spatial scale using vegetation indices generated from satellite imagery in a process called Object-Based Crop Identification and Mapping (OCIM) [57]. Similarly, they used thresholds for each index or selected textural feature that separated each field into different groups until eventually reaching a conclusion about the crop type in that field. The results here are further confirmation of the ability to delineate crop sub-populations using objective categorical groupings of continuous quantitative variables.

In previous work, NDRE has been shown to be sensitive to N uptake in maize [27,58]. Although the CI explained more variation in grain yield at the experimental site (Table 6), for other crop productivity parameters and under more generalized conditions (Table 7), NDRE provided similar or superior information to the CI on its own. Therefore, without the presence of water stress, using the CI may overcomplicate or even hinder analyses compared to using NDRE$_N$ on its own. In management systems where water stress is likely, or in research settings where the effects of irrigation and water treatments are being measured, the combination of NDRE$_N$, GLI$_N$, and BRI$_N$ in the form of the CI could be useful for differentiating N and water stress signals. However, when crop water stress is unlikely or uncertain, NDRE may more reliably identify crop stress. Individual pixel analysis indicated that the NDRE was less sensitive to the environmental and management differences between the experimental sites, the validation sites, and even between the mineral and organic N treatments within validation site 2. This indicates that NDRE may be useful for identifying stress in varying management scenarios and may not be as sensitive to in-field variation as blue or GLI, making it a more robust indicator of stress across environments and management practices.

For both NDRE and the CI, crop stress was more clearly detectable during the reproductive stages of growth than during the vegetative stages. Early vegetative stages are generally when N management decisions will have the greatest impact on crop productivity outcomes. Therefore,

stress signals detectable during the vegetative growth period would be more useful for indicating potential corrective management actions than signals that are not detectable until the reproductive growth stages. Nevertheless, at R2, approximately 19% of seasonal N uptake remained (Figure 3), so N applications to a deficient crop could still theoretically impact productivity at this stage. Along similar lines, yield responses to late-season N applications have been reported in maize, but depend on the degree of N-deficiency and the plasticity of the variety [59–61].

## 5. Conclusions

The results of this study showed that a categorically-derived CI consisting of a decision tree schematic with $NDRE_N$, $GLI_N$ and $BRI_N$ can differentiate between nitrogen and water stress only in an environment where the two forms of stress exist. $NDRE_N$ alone had a strong relationship with crop productivity across stress gradients regardless of management and can be used to indicate stress across environments. At the experimental site, the treatment structure had imposed nitrogen and water deficiency that led to differentiation in terms of total biomass, total N uptake, and final yields. In that case, the CI successfully identified all extreme deficiency. Since the $GLI_N$ was sensitive to varying rates of maturity in the nitrogen deficient treatments, the CI became more precise at the later growth stage, and timing of measurement was critical in the utility of the CI. At the validation sites, since there was no water stress, the full CI was not useful and $NDRE_N$ alone had a strong relationship with yields and was better able to explain stress, regardless of management. Since the $NDRE_N$ explained just as much variation in final yields as the CI at the experimental site, the $NDRE_N$ alone is a simpler index to use to assess stress across all the management systems explored in this study. These results show that reflectance indices can be used to indicate stress either on a scale of severity or to differentiate between types of stress where applicable. Further studies based on this work may explore differentiating between nitrogen and water stress in a potentially co-limited environment.

**Author Contributions:** Conceptualization, M.E.L. and M.L.-M.; methodology, M.E.L. and T.B.; software, T.S.N. and T.B.; validation, T.B. and M.E.L.; formal analysis, T.B.; investigation, T.B.; resources, M.E.L.; data curation, T.B.; writing—original draft preparation, T.B.; writing—review and editing, M.E.L., M.L.-M., T.S.N.; visualization, M.E.L.; supervision, M.E.L.; project administration, M.E.L.; funding acquisition, M.E.L. All authors have read and agreed to the published version of the manuscript.

**Funding:** This research was funded by the Department of Plant Sciences, University of California, Davis and the University of California Division of Agriculture and Natural Resources.

**Acknowledgments:** Gary and Steve Mello, Israel Herrera and Russell Ranch Field Crew, Nicole Tautges, Luis Loza, Ethan McCullough, Michael Rodriguez, Kalyn Diederich.

**Conflicts of Interest:** The authors declare no conflict of interest. The funders had no role in the design of the study; in the collection, analyses, or interpretation of data; in the writing of the manuscript, or in the decision to publish the results.

# Appendix A

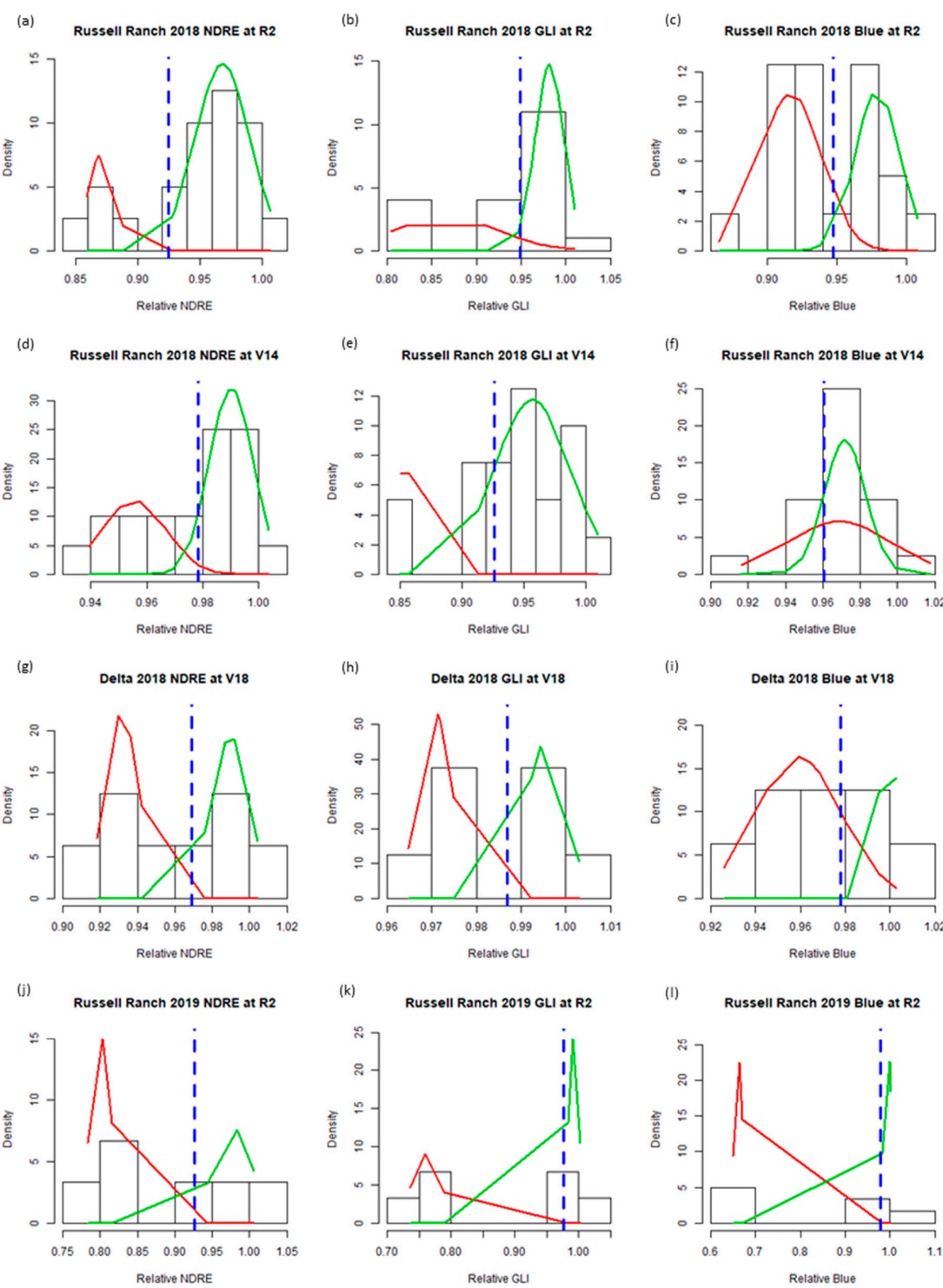

**Figure A1.** Density plot results from Gaussian mixture modeling analysis of the normalized difference red-edge (NDRE), green leaf index (GLI), and blue reflectance values for (**a**–**c**) the experimental site at blister (R2) and (**d**–**f**) the 14th leaf stage (V14), (**g**–**i**) validation site 1 at the 18th leaf stage (V18), and (**j**–**l**) validation site 2 at R2. Red and green curves show the modeled distribution of data split into each group by mixture modeling. The blue dotted line shows the thresholds used to split data into the two groups, with data below the threshold but in the "low" group and data above the threshold put into the "high" group.

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
