# Peer review of "Differentiating between Nitrogen and Water Deficiency in Irrigated Maize Using a UAV-Based Multi-Spectral Camera"

_agronomy, doi:10.3390/agronomy10111671_

Round 1
Reviewer 1 Report
The authors have addressed all my comments/suggestions raised in my previous review.
Reviewer 2 Report
The authors' responses are good.
This manuscript is a resubmission of an earlier submission. The following is a list of the peer review reports and author responses from that submission.
Round 1
Reviewer 1 Report
The manuscript entitles “Differentiating between nitrogen and water deficiency in irrigated maize using a UAV-based Multi-spectral camera” (Manuscript#: agronomy-898266, Article) has been reviewed for publication in Agronomy. The authors tried to analyze the differentiation of water deficiency and nitrogen deficiency in maze by multispectral camera.
I request the biological explanation for readers who are unfamiliar to maize research.
(1) The maize growth stages mentioned in this manuscript are essential, V7, V10, V17 and R2 and so on. Although the authors explain in this manuscript, please show the stages, figures/images in the supplementary figure or table for readers.
(2) The names of maize cultivars (NuTech OA713 and Limagrain ES7514) are also unfamiliar. Are these F1 hybrids? Please indicate the characteristics in detail.
(3) Please unify maize instead of corn.
How do the authors calculate CI grouping successful rate (from Line 366)?
Reviewer 2 Report
The authors conducted very interesting research in an attempt to discriminate between nitrogen and water stress in irrigated maize by multispectral observation, assessing single and combined vegetation indexes.
The authors provide data and observations that may be of great interest, unfortunately the structure of this manuscript is poor and this often makes it difficult to understand the methodology and the results.
English is perfect and this allowed me to come out from the confused M&M and Results section, but it won't be enought for a young researcher or readers.
What's good about your manuscript? The discussion, except for LL467-476, is excellent. Figures are clear too.
The introduction should be improved by giving more space to other index (Ipca for example) and then to justify your choice (GLI, B adn NDRE)
What needs to be deeply revised?
M&M must be splitted and re-organized into sub-chapters (one for each activity), following a clear mind map.
The whole part relating to the "modeling" and statistical analysis should be reorganized in a sort of conceptual framework so as to make clear the part of "Selection of Sensitive Indices" in Results
Reorganizing M&M in a clear and linear way will certainly make it possible to reorganize the results in the best possible way.
Eventually I'm sure the authors will find a better way to rewrite the abstract as well.
minor
L18 please provide stage name and code btw brackets (throughout the manuscript)
L32-L33 please provide a recent ref
L63 please introduce briefly remote sensing (before you are reporting about proximal sensing)
L92-95 could be deleted, please focus on other index that you do not consider in your study to better justify the adoption of GLI, NDRE and B instead than others.
L143 any ref for CIMIS?
Table1 please check if it is correct reporting Pre-plant soil N value also for Delta2018 no preplant N plot. If these values refer to N content in the soil, please clarify that you analyzed one sample per site
L310 please report the name of your experimental site...at this point it has already been forgotten
L467-L476 please delete "Under a narrow set of circumstances" and use the rest for your introduction.